# Study on the Phase Transition from Quartz to Coesite under High Temperature and High Pressure

Dongsheng Ren

State Key Laboratory of Earthquake Dynamics, Institute of Geology, China Earthquake Administration, Beijing 100029, China; rendongsheng@ies.ac.cn

**Abstract:** Quartz is an important component of the Earth. In this study, experiments were conducted at temperatures between 600 to 700 °C, confining pressures between 1.5 and 1.8 GPa, and differential stress conditions. It was found that coesite production is closely related to differential stress, reaction time, and reaction temperature, with coesite formation being a multifactorial coupling process.

**Keywords:** high temperature; high pressure; differential stress; coesite; phase transition





## 1. Introduction

Quartz is an important component of the Earth's interior, and the study of its phase transitions under high pressure is crucial to understanding the stability of various mineral assemblages, and the temperature and pressure environment deep in the Earth's interior, and to determining the diagenesis depth of certain rocks containing high-pressure mineral inclusions. In laboratory studies, the quartz phase transitions to coesite, then stishovite at high temperatures and pressures, and the super-stishovite phase has a $CaCl_2$-type and $\alpha$-$PbO_2$-type structure under ultra-high pressure [1–3].

Since the temperature and pressure conditions required for the phase transition of quartz into coesite are equivalent to those in the Earth's interior at a depth of about 100 km, the coesite found near the Earth's surface may have been brought to the surface by geological activity after forming at such depths. In addition to the above foldback hypothesis [4], the formation of high-pressure quartz phases may also be caused by shock and differential stress effects [5–9]. It has been demonstrated that coesite can occur at a confining pressure of 1 to 2 GPa under conditions of differential stress and large strain [10,11]. Hirth and Tullis (1994) investigated the brittle–plastic transformation of quartz using a piston-cylinder pressure apparatus and found that stable coesite is observed at low confining pressures (1.2 to 1.25 GPa) in the presence of relatively large differential stresses and medium strains [12]. The same conclusion was reached in high-temperature and high-pressure experiments [1]. Richter (2016) [13] identified coesite formation by shear experiments at 600 to 900 °C and 1 to 1.5 GPa. These studies indicate that coesite formation under differential stress conditions requires a lower confining pressure than what is necessary to stabilize coesite under hydrostatic pressure (>3 GPa) [13]. However, the above studies focused on the important role of the maximum principal stress. They believe that coesite can appear if the maximum principal stress exceeds the phase transition pressure while being under hydrostatic pressure at the same temperature.

In this study, we conducted experiments at 600 to 700 °C and 1.5 to 1.8 GPa using the Griggs apparatus to investigate the relationship between the quartz–coesite phase transition and temperature, phase transition time, strain, and differential stress, which is of great significance.

## 2. Experimental Materials and Methods

### 2.1. Starting Materials

The α-quartz powder used in these experiments was purchased from Aladdin Co., Ltd. in Shanghai, China. It had a density of 2.65 g/cm$^3$, a purity of 99%, and a particle size of 250 μm. The α-quartz was dried thoroughly in an oven at 150 °C. A sample of 0.850 g of powder was weighed on an analytical balance and wrapped with silver foil, then placed in a φ10 press mold and pressed to 2 MPa using a powder pressing machine to obtain a cylinder with a length of 20 ± 0.05 mm and a diameter of 10 ± 0.05 mm.

### 2.2. Experimental Design and Process

The sample assembly of the experiment is shown in Figure 1; this machine is in the Institute of Geology, China Earthquake Administration [1]. The axial pressure of this equipment can be controlled by load and displacement. First, we controlled the load by applying a small pressure (about 0.5 GPa) to the sample, as to make the components in the sample have close contact. At a lower pressure, we heated the sample to the target experimental temperature at a heating rate of 10 °C/min. Then, we continued to use the load control method to load the pressure on the sample to the target confining pressure. Finally, we controlled the axial pressure through displacement loading with the loading speed of $5 \times 10^{-6}$ m/s; after loading to the target pressure value, the pressure was maintained. The schematic diagram of pressure loading and heating process is shown in Figure 2. The value of the confining pressure was obtained through pressure calibration, the axial pressure was calculated according to the axial pressure load, and the strain was calculated through the axial displacement sensor. After the experiment, the heating power supply was cut off, and the axial pressure and confining pressure were quickly unloaded to make the sample quench quickly. Then, the samples were removed and analyzed by Raman spectroscopy to determine if any phase transitions had occurred. Raman spectroscopy was completed at the Institute of Geochemistry, Chinese Academy of Sciences. For details, see [14]. The samples used for the Raman spectrum test were selected from the products after the experiment, and the specific position was selected on the upper surface, close to the corundum column (No. 7 in Figure 1). The reason for choosing this place is that the differential stress environment here is conducive to the production of coesite. The pressure environments of other parts of the sample consist of quasi-hydrostatic pressure. If coesite does not appear at the selected test location, it is almost impossible to appear elsewhere.

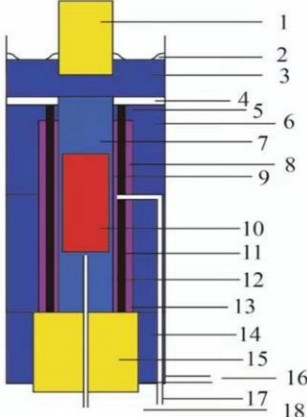

**Figure 1.** The α-quartz phase transition experimental assembly. (1) Tungsten carbide shaft rod, (2) confining pressure piston sealing ring, (3) lead ring, (4) copper gasket, (5) upper conductive copper ring, (6) pressure-transmitting medium salt sleeve, (7) corundum column, (8) outer pyrophyllite sleeve, (9) sample sealing sleeve, (10) sample, (11) graphite furnace, (12) inner pyrophyllite sleeve, (13) lower conductive copper ring, (14) lower pyrophyllite sleeve, (15) tungsten carbide briquette, (16) insulating mica sheet, (17) thermocouple, (18) sample bottom thermocouple.

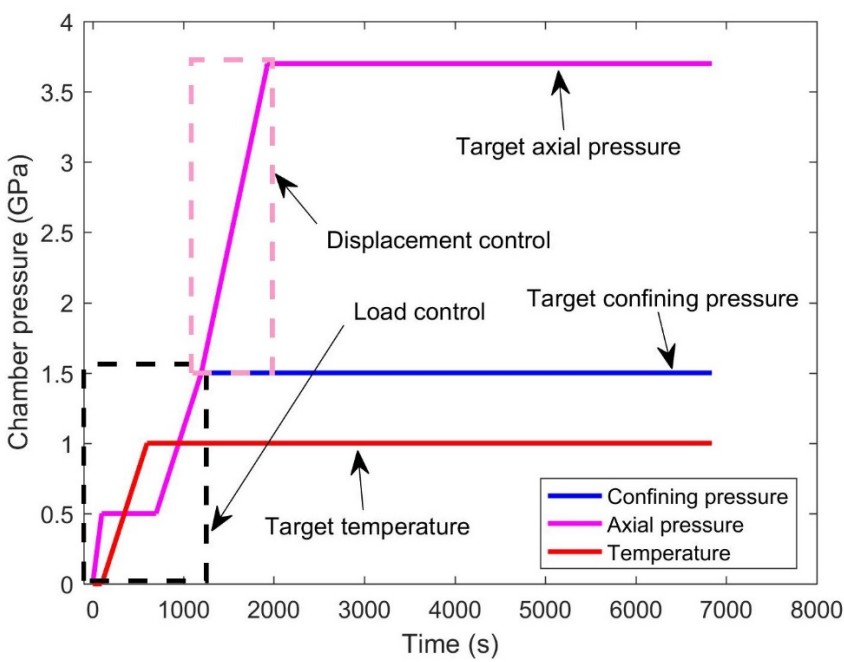

**Figure 2.** Pressure loading and heating process during the experiment.

## 3. Results and Discussion

A total of five sets of experiments were conducted in this study, as shown in Table 1. The resulting samples were sectioned for Raman spectroscopy analysis. Experiments 1 and 3 did not reveal coesite being formed, whereas Experiments 2, 4, and 5 found coesite at the corundum indenter. Figure 3 shows the Raman spectra of the starting materials and the resulting products from Experiments 2, 4, and 5.

**Table 1.** Experimental conditions of the quartz phase transition under differential stress.

| | Temperature/°C | Confining Pressure/GPa | Maximum Principal Stress/GPa | Differential Stress/GPa | Mean Stress/GPa | Strain | Holding Time/h | Coesite Present (Y/N) |
|---|---|---|---|---|---|---|---|---|
| 1 | 700 | 1.5 | 3.7 | 2.2 | 2.0 | 72.1 | 36 | N |
| 2 | 700 | 1.6 | 3.9 | 2.3 | 2.1 | 72.3 | 36 | Y |
| 3 | 600 | 1.8 | 4.1 | 2.3 | 2.4 | 73.8 | 24 | N |
| 4 | 600 | 1.8 | 4.2 | 2.4 | 2.4 | 75.6 | 36 | Y |
| 5 | 700 | 1.7 | 4.0 | 2.3 | 2.3 | 71.1 | 24 | Y |

The experimental data presented in Table 1 and previous experimental data obtained under similar experimental conditions are plotted on the quartz–coesite phase transition diagram under hydrostatic pressure (Figure 4) [1,11]. As observed, the results from this experimental study are close to those of Hirth (1994) [12] and differ somewhat from those of Zhou et al. (2005) [1]. In Zhou et al. (2005) [1], except for the cultellation of the maximum principal stress, which is close to the quartz–coesite phase transition line under hydrostatic pressure, the cultellation of the confining pressure, differential stress, and average stress were much lower than the quartz–coesite phase transition line under hydrostatic pressure; the experimental temperature was about 1000 °C, which is very high. According to the current study and Hirth's findings, only the cultellation of the confining pressure is much lower than the quartz–coesite phase transition line under hydrostatic pressure, as the cultellation of the maximum principal stress is much higher than the quartz–coesite phase transition line under hydrostatic pressure, and the cultellation of the differential stress and average stress is distributed near the quartz–coesite phase transition line. However, the reaction temperatures in this study were 600 and 700 °C, which is relatively low.

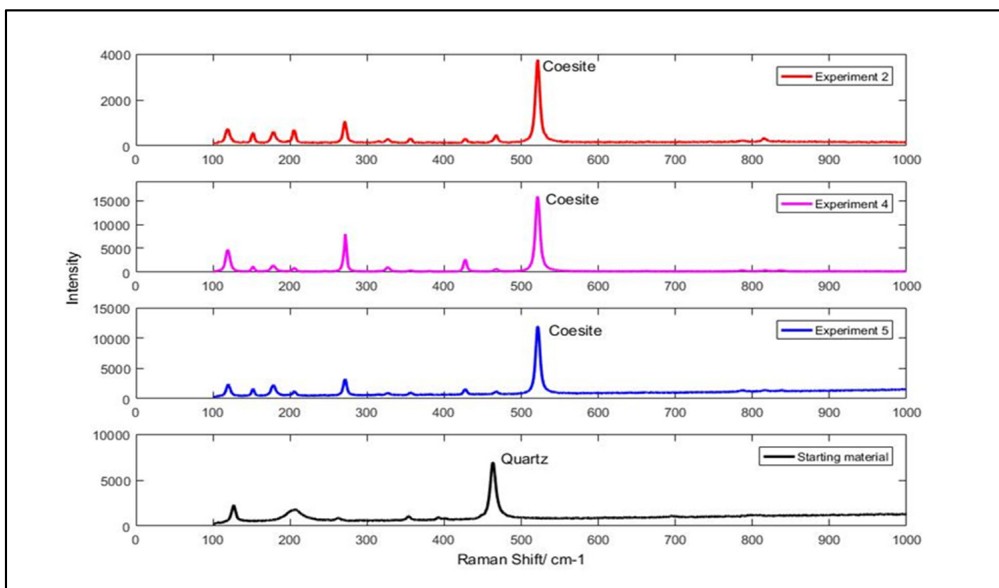

**Figure 3.** Raman spectra of the raw material and the products after the experiments that resulted in coesite forming.

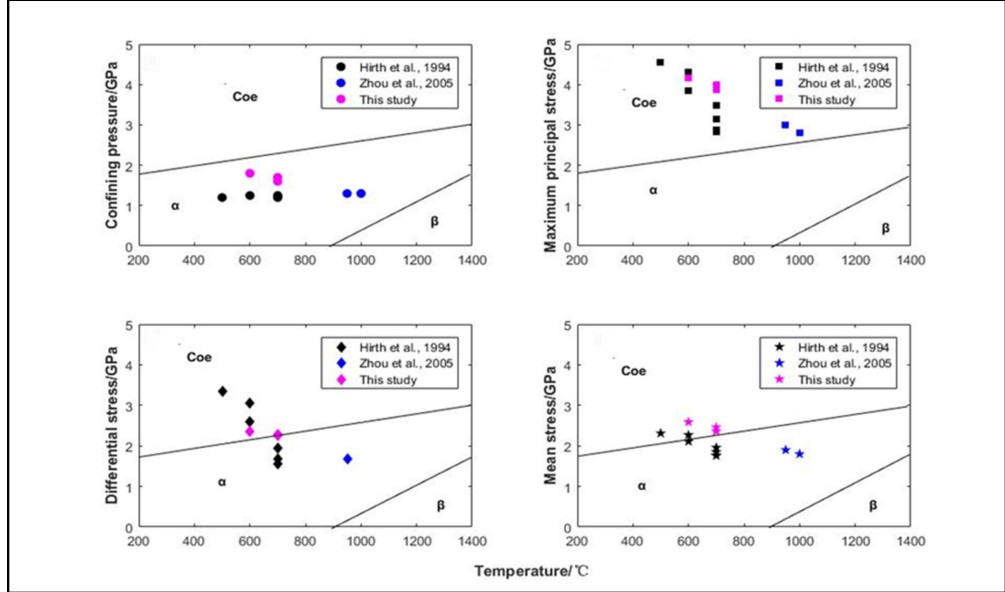

**Figure 4.** Pressure and stress conditions as a function of temperature for different phases of quartz. Coe—coesite [1,12].

For Experiments 1 and 2, the reaction temperature and pressure holding time are the same, but the maximum principal stress and differential stress in Experiment 2 are greater than those in Experiment 1. No coesite developed in Experiment 1, but it did develop in Experiment 2. Therefore, it is believed that the maximum principal stress and differential stress are the key factors behind coesite formation in this case. This view is also consistent with that of Hirth (1994) [12]. Meanwhile, we found that the coesite production is also related to the pressure holding time, and a shorter pressure holding time is not conducive to the production of coesite, which is derived from comparing the results of Experiments 3 and 4. In these experiments, the reaction temperature and stress conditions are similar but the pressure holding time differs, with no coesite developing in Experiment 3, although it did develop in Experiment 4. In addition, we also found that the higher temperature in this deformation region favors the emergence of coesite, which we concluded from

comparing the results of Experiments 3 and 5. The comparison between the experimental results of Zhou et al. (2005) [1] and our results and those of Hirth (1994) [12] also shows the temperature effect on coesite formation.

In Zhou et al. (2005) [1], the large deformation is also a major factor in coesite generation. In this study, the quartz deformation is also very large, with the strain exceeding 70%; thus, we believe that a large deformation is one of the main reasons for the development of coesite.

All of the studies mentioned above (Zhou et al., 2005 [1]; Hirth, 1994 [12]; and this study) indicate that the phase transformation of quartz to coesite can occur at a certain temperature with a small sample confining pressure and high differential stress. However, it is believed that the emergence of coesite is a complex process, which is not dominated by a single factor but involves multifactorial coupling, and that the quartz–coesite phase transition is only observed at a specific temperature (500 °C) and under high differential stress with a long reaction time.

## 4. Conclusions

In this study, experiments were conducted over the temperature range of 600 to 700 °C, confining pressures of 1.5 to 1.8 GPa, and differential stress conditions. It was found that the generation of coesite is closely related to differential stress, reaction time, and reaction temperature, with coesite formation being the result of a multifactorial coupling process. Compared with previous studies (Hirth, 1994 [12]; Zhou et al., 2005 [1]), the highlight of our experiment is the discovery of coesite under low confining pressure and large strain conditions. However, there are some limitations in these results. Current studies have been restricted to the qualitative discussion of coesite formation conditions, but no quantitative experimental results are available to help explain the relationships between coesite formation and differential stress.

**Funding:** This research was funded by the National Natural Science Foundation of China, grant number 42104176.

**Data Availability Statement:** Not applicable.

**Conflicts of Interest:** The author declares no conflict of interest.

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
