# Peer review of "Study on the Phase Transition from Quartz to Coesite under High Temperature and High Pressure"

_minerals, doi:10.3390/min12080963_

Round 1

Reviewer 1 Report

Quartz is an important component of the Earth, and the study of its phase transitions under high pressure is crucial to understanding the stability of various mineral assemblages in the Earth's interior, the temperature and pressure environment in the deep Earth, and determining the diagenesis depth of certain rocks containing high-pressure mineral inclusions. To sum up, this paper is very meaningful.  But I have some questions and suggestions, which I hope will be helpful to the author.
1. During the experiment, the samples are subjected to large differential stress and undergo large deformation. Is this inconsistent with the actual geological process?
2. It is mentioned that reaction time, differential stress and maximum principal stress all contribute to the formation of coesite. Which is the most important factor?
3. HT-HP can be used to replace high temperature and high pressure in this paper.

Author Response

Thank you for your comments and replies. I will reply to your questions as below.

Q1. During the experiment, the samples are subjected to large differential stress and undergo large deformation. Is this inconsistent with the actual geological process?

A1:The differential stress in nature is generally hundreds of MPA, but the nature has generally experienced a long geological process, and the laboratory needs a large differential stress to produce coesite under low confining pressure.

Q2. It is mentioned that reaction time, differential stress and maximum principal stress all contribute to the formation of coesite. Which is the most important factor?

A2: Experiments show that the maximum principal stress is the most important factor.

Q3. HT-HP can be used to replace high temperature and high pressure in this paper.

A3:This aspect has been modified

Reviewer 2 Report

Ren-Coesite-2022

This short paper touches on an interesting subject: coesite and experimental conditions. It is a topic that has received a lot of attention. Reading the paper it is not quite sure what this paper adds to our understanding. It is very brief and many issues, e.g. experimental, need to be expanded. For example Fig. 3 is puzzling for this reviewer: It is exactly the same phase diagram for different parameters like pressure, principle stress and average stress. Sometimes all data are in the quartz field and sometimes in the coesite field and distributions from different authors are very similar. Also: you start with a quartz powder. You would think stress for transformation would also be linked to crystal orientation. I guess Raman is not in situ. You should at least show an optical image of the transformed sample. Better would be SEM and especially EBSD. And also an X-ray diffraction pattern. Below are some details that need to be considered if you decide to prepare a revised version.

Details

Abstract: This short paper presents some experimental results but without any implications or references for natural coesite localities. Delete lines 8-11 of abstract.

Line 12 delete of

Line 23 here give some references to coesite in the Earth, like Dora Maira in Alps, ophiolites in Tibet, Harz-Germany, Vredefort crater, Ries impact crater, Coconino meteor crater, Wabar Crater etc. Coesite is interesting but not “crucial” for understanding the Earth’s interior.

Line 45 Can you give a reference for “Griggs apparatus”? And where is the Griggs apparatus where the experiments described in section 2.2 were conducted?

Line 50: who is Aladdin? So the sample is a fine-grained powder. Was it still a powder after the experiment?

Figure 1: give a scale

Line 79 You mention Raman spectroscopy but give no details. You say “samples were sectioned”. So they were no longer powders? You should show an optical picture of the sectioned sample. What is the size of the Raman beam? And why did you not also take an X-ray diffraction spectrum?

Table 1: “Coesite present” How much compared with quartz? And strain: Your powder was shortened 72%?

Figure 2: Why is the “raw material” at the bottom? And here it appears that all quartz in the powder transformed to coesite. And was Raman spectroscopy conducted in situ? From the text it appears that it was not.

Figure 3: Looking at this figure one wonders what is new about your study? It seems this has all been done before and was reported. And in the phase diagrams the experimental results are in top left all in the alpha field, in top right all in the coesite field and in the other two they were in between. What were the actual phases? Give different symbols to different phases. And why are all phase diagrams identical?

Reference 3 is strange give the correct year and volume!

Ref. 6, 7, 8 use capitals in titles

Author Response

First of all, thank you for your comments. Your comments have greatly improved my article. The significance of this paper is that coesite has been found in the laboratory under the condition of differential stress. The previous experimental results focus on the importance of differential stress and ignore the phase transition time. This paper emphasizes the importance of phase transition time. The Raman results in this paper are obtained from the samples after the experiment. This paper aims to emphasize that coesite is found in the experiment without discussing the adaptive deformation mechanism. Therefore, the Raman structure is sufficient to prove the production of coesite without EBSD and other analysis and testing techniques. The study of quartz deformation mechanism will be mainly shown in the future work. I will answer and improve my questions in detail as fellow.

Q1:Abstract: This short paper presents some experimental results but without any implications or references for natural coesite localities. Delete lines 8-11 of abstract.

A1:You are right. This part has been deleted.

Q2: Line 23 here give some references to coesite in the Earth, like Dora Maira in Alps, ophiolites in Tibet, Harz-Germany, Vredefort crater, Ries impact crater, Coconino meteor crater, Wabar Crater etc. Coesite is interesting but not “crucial” for understanding the Earth’s interior.

A2: Your statement is correct. The literature is added here.

Q3: Line 45 Can you give a reference for “Griggs apparatus”? And where is the Griggs apparatus where the experiments described in section 2.2 were conducted?

A3: This part of references has been added.

Q4: Line 50: who is Aladdin? So the sample is a fine-grained powder. Was it still a powder after the experiment?

A4: The description of this section has been changed. The sample has been pre pressed and is no longer a powder, but it's not tight enough.

Q5: Figure 1: give a scale

A5: Scale bar has been added

Q6:Line 79 You mention Raman spectroscopy but give no details. You say “samples were sectioned”. So they were no longer powders? You should show an optical picture of the sectioned sample. What is the size of the Raman beam? And why did you not also take an X-ray diffraction spectrum?

A6: The specific description has been added, and the sample has been cut into thin slices. The selected position is near the indenter, because the differential stress here is relatively large, it is easy to find coesite. Because our purpose is to check whether coesite is produced, Raman spectrum is the same as XRD.

Q7: Table 1: “Coesite present” How much compared with quartz? And strain: Your powder was shortened 72%?

A7: No quantitative research has been done, but the total amount of coesite should be less than 10%. The main reason why the strain reaches 70% is that the pre pressed samples are relatively loose and undergo high differential stress extrusion.

Q8: Figure 2: Why is the “raw material” at the bottom? And here it appears that all quartz in the powder transformed to coesite. And was Raman spectroscopy conducted in situ? From the text it appears that it was not.

A8:  Coesite was found in Experiments 2, 4 and 5. Through Raman spectrum analysis, the main peak of coesite was indeed found in the test part selected in Experiments 2, 4 and 5. Placing the initial substance at the bottom also confirmed that the main peak of the product after the experiment was indeed shifted from that before the experiment. This is not in situ Raman.

Q9: Figure 3: Looking at this figure one wonders what is new about your study? It seems this has all been done before and was reported. And in the phase diagrams the experimental results are in top left all in the alpha field, in top right all in the coesite field and in the other two they were in between. What were the actual phases? Give different symbols to different phases. And why are all phase diagrams identical?

A9: The pink data are our results. Our experimental results are similar to the previous findings, and there are also new findings. The similarity is that coesite may be found if the maximum principal stress exceeds the phase change pressure of hydrostatic pressure at the same temperature. However, we found that a longer phase transition time is conducive to the production of coesite, which is rarely mentioned by predecessors. See reference 1 for drawing method.

Q10: Reference 3 is strange give the correct year and volume!

A10: These have been modified

Q11: Ref. 6, 7, 8 use capitals in titles

A11: These have been modified

Reviewer 3 Report

The manuscript, Study on the phase transition from quartz to coesite under high temperature and high pressure, submitted by Ren reports high-pressure temperature experiments on quartz to coesite transition under various conditions. The results qualitatively indicate the formation of coesite relates to differential stress, reaction time, and reaction temperature. In general, this is an incomplete study, and this manuscript should be rejected for the following reasons:

  1. Experimental methods are unclear. How were confining pressure, maximum principal stress, and strain determined, respectively? What are the uncertainties of these variables? Lines 60-61 about temperature descriptions are confusing. No descriptions of Raman spectroscopy measurements.
  2. The characterizations of experimental products are incomplete. The author only provides three Raman spectra for 3 experiments, and no data at all for the rest of 2 experiments. The reported data is not enough to support no coesite observed in #1 and #3 experiments. Powder XRD patterns for all experimental products are necessary.
  3. Given the unclear experimental methods and results, the discussions are premature. Nevertheless, the manuscript doesn’t provide any new insight on quartz to coesite transition.

Author Response

First of all, thank you for your comments. The significance of this paper lies in the discovery of coesite under laboratory conditions. Compared with the previous experimental results, our samples experienced large strain, low confining pressure, and low experimental temperature. At the same time, we emphasize the importance of phase transition time. Therefore, the highlight of this paper is that coesite was found under the condition of large strain and low temperature without shear. For example, Zhou (2005) et al. Studied that the experimental temperature reached 1000 ℃, green (1972) studied that the strain was very small, and Richter (2016) studied the shear deformation experiment. Our experiment is a pure compression experiment, which complements and perfects the relevant research. Coesite was found under the condition of large strain and low temperature. The paper has been carefully revised, and the specific reply is as follows.

Q1: Experimental methods are unclear. How were confining pressure, maximum principal stress, and strain determined, respectively? What are the uncertainties of these variables? Lines 60-61 about temperature descriptions are confusing. No descriptions of Raman spectroscopy measurements.

A1: The value of confining pressure is obtained through pressure calibration, the axial pressure can be calculated according to the axial pressure load, and the strain can be calculated through the axial displacement sensor. This part has been modified in the text.

Q2: The characterizations of experimental products are incomplete. The author only provides three Raman spectra for 3 experiments, and no data at all for the rest of 2 experiments. The reported data is not enough to support no coesite observed in #1 and #3 experiments. Powder XRD patterns for all experimental products are necessary.

A2. The experiment without coesite was also tested by Raman spectroscopy, but coesite was not found on the same surface. Our goal is to confirm that coesite can be produced under the condition of large strain, low temperature and differential stress, and the Raman spectrum test is sufficient to confirm it. The selected position is representative. If there is no coesite on this face, there will be no coesite in other positions.

Round 2

Reviewer 2 Report

Manuscript has been substantially improved

Author Response

Thank you for your comments, which has greatly improved my paper.

Reviewer 3 Report

I do not think this manuscript is publishable.

Author Response

Compared with previous studies, the highlight of this paper is that coesite was found under the condition of large strain and low temperature without shear. For example, Zhou (2005) et al. Studied that the experimental temperature reached 1000 ℃, green (1972) studied that the strain was very small, and Richter (2016) studied the shear deformation experiment. Our experiment is a pure compression experiment, which complements and perfects the relevant research. Coesite was found under the condition of large strain and low temperature. The paper has been carefully revised, and the specific reply is as follows.

Q1: Experimental methods are unclear. How were confining pressure, maximum principal stress, and strain determined, respectively? What are the uncertainties of these variables? Lines 60-61 about temperature descriptions are confusing. No descriptions of Raman spectroscopy measurements.

A1: The value of confining pressure is obtained through pressure calibration, the axial pressure can be calculated according to the axial pressure load, and the strain can be calculated through the axial displacement sensor. To facilitate understanding, we have added a pattern diagram.

Q2: The characterizations of experimental products are incomplete. The author only provides three Raman spectra for 3 experiments, and no data at all for the rest of 2 experiments. The reported data is not enough to support no coesite observed in #1 and #3 experiments. Powder XRD patterns for all experimental products are necessary.

A2. Our goal is to confirm that coesite can be produced under the condition of large strain, low temperature and differential stress, and the Raman spectrum test is sufficient to confirm it. But according to your opinion, we did XRD test on the samples of Experiment 1 and Experiment 3, which were also quartz, and no coesite was found.

Q3. Given the unclear experimental methods and results, the discussions are premature. Nevertheless, the manuscript doesn’t provide any new insight on quartz to coesite transition.

A3. Compared with Hirth's experiment, under similar temperature and differential stress conditions, the strain of our experimental samples reached 70%, much higher than their 20%. As a result, coesite was found in our experiment, but they did not. Therefore, we can see that large strain is conducive to the production of coesite. This is a relatively innovative understanding.